# A Novel Mechanical Fault Feature Selection and Diagnosis Approach for High-Voltage Circuit Breakers Using Features Extracted without Signal Processing

**DOI:** 10.3390/s19020288

**Published:** 2019-01-12

**Authors:** Lin Lin, Bin Wang, Jiajin Qi, Lingling Chen, Nantian Huang

**Affiliations:** 1College of Information and Control Engineering, Jilin Institute of Chemical Technology, Jilin 132022, China; cll807900@163.com; 2Taian Power Supply Company, State Grid Shandong Electric Power Company, Taian 271000, China; taian_wangbin@126.com; 3Hangzhou Municipal Electric Power Supply Company of State Grid, Hangzhou 310009, China; qijiajin@126.com; 4School of Electrical Engineering, Northeast Electric Power University, Jilin 132012, China; huangnantian@neepu.edu.cn

**Keywords:** high voltage circuit breaker, one-class support vector machine, random forest

## Abstract

The reliability and performance of high-voltage circuit breakers (HVCBs) will directly affect the safety and stability of the power system itself, and mechanical failures of HVCBs are one of the important factors affecting the reliability of circuit breakers. Moreover, the existing fault diagnosis methods for circuit breakers are complex and inefficient in feature extraction. To improve the efficiency of feature extraction, a novel mechanical fault feature selection and diagnosis approach for high-voltage circuit breakers, using features extracted without signal processing is proposed. Firstly, the vibration signal of the HVCBs’ operating system, which collects the amplitudes of signals from normal vibration signals, is segmented by a time scale, and obviously changed. Adopting the ensemble learning method, features were extracted from each part of the divided signal, and used for constructing a vector. The Gini importance of features is obtained by random forest (RF), and the feature is ranked by the features’ importance index. After that, sequential forward selection (SFS) is applied to determine the optimal subset, while the regularized Fisher’s criterion (RFC) is used to analyze the classification ability. Then, the optimal subset is input to the hierarchical hybrid classifier, and based on a one-class support vector machine (OCSVM) and RF for fault diagnosis, the state is accurately recognized by OCSVM. The known fault types are identified using RF, and the identification results are calibrated with OCSVM of a particular fault type. The experimental proves that the new method has high feature extraction efficiency and recognition accuracy by the measured HVCBs vibration signal, while the unknown fault type data of the untrained samples is effectively identified.

## 1. Introduction

High-voltage circuit breakers (HVCBs) are the most important control and primary protection equipment in electric power systems [1,2]. The operation states of HVCBs are directly related to the stability and reliability of the power system. The analysis object of HVCBs mainly operates via the moving contact travel-time characteristic method, the tripping (closing) coil current method, and the vibration signal method at present [3,4,5,6]. The operating mechanism is the main factor affecting the reliability of circuit breakers. There are many mechanical failures, such as a lack of spring energy storage, and screw loosening by vibration signals [7]. Therefore, HVCBs fault diagnoses based on vibration signals is of great significance [8]. The analysis process mainly includes signal processing, feature extraction, and fault diagnosis.

Due to the non-stationary and the nonlinear characteristics of the vibration signals of HVCBs, traditional signal processing methods analyze the vibration signal in the time domain and frequency domain, and extract time–frequency domain features. The commonly used signal processing methods are empirical mode decomposition (EMD) [9,10], ensemble empirical mode decomposition (EEMD) [11], and local mean decomposition (LMD) [12], etc. The above methods have achieved good fault diagnosis results, but there are still some shortcomings. Feature extractions with EMD and LMD have some problems in the process of decomposition, such as mode mixing and end effects [11]. Although EEMD has suppressed mode mixing by adding white noise, the method also increases the amount of computation and decomposes many components beyond the real composition of the signal [12]. At the same time, the process of the signal processing method is complex, with high time complexity, which improves the computational cost and the industrialization difficulty of the related technologies.

The features of the traditional vibration signals include time domain features, frequency domain features, and time-frequency features. The existing features set can accurately describe the different state of the fault signals. However, the features of HVCBs vibration signals are widely distributed in the frequency and time domains, and it is difficult to extract effective features from the specific frequency domain or the time domain, which are affected by the specific installation environment in the actual work [13]. Because of the differences in the degree of attenuation and starting time of the different types of fault vibration signals, HVCBs failure states can be analyzed by the time domain features when directly extracted from the original vibration signal. The features are extracted by calculating the mean, variance, and standard deviation by different time domain segmentation scales from the original signal [13,14,15,16]. By extracting the abundant original signal features, the fault information can be described in detail. However, the dimensions of the feature set will be increased, and redundant features may be included in the feature set. The feature set with high dimensional features will affect the fault diagnosis accuracy and efficiency of the classifier. Therefore, selecting the optimal feature subset from higher-dimensional features extracted from original signals is the key to improve the efficiency and accuracy of HVCBs fault diagnosis.

Fault diagnosis methods for circuit breakers include support vector machine (SVM) [17], neural networks (NNs) [7], etc. However, there are many kinds of mechanical faults in HVCBs, and the operation of HVCBs is rare. Furthermore, the cost of obtaining the fault sample experiment is high. It is difficult to acquire enough fault samples with all fault types. Traditional multi-classifiers are easily identify the fault type data without training samples as the known or normal states. The effect of condition monitoring is seriously affected.

To improve the efficiency of feature extraction of vibration signals, and to avoid untrained samples of unknown type faults from being identified as normal or error known types, a new method of mechanical fault diagnosis for HVCBs based on feature extraction and selection without signal processing is proposed. Firstly, the vibration signals of the HVCBs are segmented by a time scale that starts collecting standard normal vibration signals. Secondly, time–domain features are extracted from each part of the divided signal and used to construct the feature vector. The sequential forward selection method with the regulated Fisher’s criterion (RFC) index is used to determine the optimal feature subset, based on Gini importance. Finally, the optimal subset construction of the hierarchical hybrid classifier is based on the one-class support vector machine (OCSVM) and random forest (RF) for state recognition. The effectiveness of the new method is verified by the measured signal.

## 2. Feature Importance and Fault Classifiers

### 2.1. Gini Importance

The Gini importance is used to measure the node impurity, and it can be used to measure the feature importance [18]. Suppose that *S* is a dataset containing *s* samples, which can be divided into *n* classes. $s_{a}$ is the number of samples contained in class *a*. The Gini index of the set is:(1)$$Gini(S)=1-\sum_{a=1}^{n}P_{a}^{2}$$
where $p_{a}=p(s_{a}/S)=s_{a}/S$, which is used to expressed the probability of any sample belonging to class *a*.

When RF uses a feature to divide nodes, it can divide *S* into *m* subsets, denoted with $S_{c}(j=1,2,\cdots ,m)$, The Gini index of split *S* is:(2)$$Gini_{split}(S)=\sum_{j=1}^{m}\frac{S_{c}}{S}Gini(S_{c})$$
where $s_{c}$ is the samples number in subset $S_{c}$.

The Gini importance is:(3)$$\Delta Gini(S)=Gini(S)-Gini_{split}(S)$$

It is known from Equation (3) that the higher the Gini importance, the better the feature division [18].

### 2.2. Random Forest

RF is composed of a series of classification and regression tree (CART) models $\left\{h(X,\theta_{l}),l=1,\cdots ,L\right\}$, and voting by multiple decision trees, where $h(X,\theta_{l})$ is a classification model for CART, *X* is an input feature vector, $\left\{\theta_{l}\right\}$ is a random vector that follows the independent and identical distribution, and *l* represents the number of the classifier. For a given independent variable *X*, the optimal classification is achieved by aggregating the voting results of each CART. The detailed classification principle of RF can be found in [19,20,21], and the basic classification process is as follows:(1)*q* samples are randomly extracted from the original set *Q*, to constitute a self-help sample set, repeated *l* times.(2)During the training process, random selection from the feature space *M* is a candidate feature of non-leaf node splitting, and the nodes are divided with each candidate feature, and the best segmentation feature is chosen as the segmentation feature of the node. This process is repeated until all of the non-leaf nodes of each tree are classified, and the training process is then ended.(3)Determining the optimal classification results by the majority voting method of each of the classification results.

The optimum ranges of the minimum leaf number (MinLS) and the candidate feature number (NumPTS) for each node are 1≤MinLS≤L and 1≤NumPTS≤T. The value of L is set to 10, and T is the dimension of the feature subset. RF integrates the characteristics of bagging and random selection feature splitting; its advantages are: (1) out-of-bag (OOB) data generated by the bagging method, and it can be used to measure the importance of a single variable and estimate the generalization error of the combined classifier models; (2) Due to the large number theorem, with the increase of decision tree in RF, it is not easy to be over-fitted; (3) The algorithm can tolerate abnormal values and noises properly, and it has high classification accuracy [22].

### 2.3. One-Class Support Vector Machine (OCSVM)

OCSVM only uses normal samples to complete the training process and to determine the mechanical state of the device. The speed of the training and decision is fast, and the anti-noise performance is good. It is suitable for the field of mechanical condition monitoring with high reliability.

Suppose there is a sample $\left\{z_{i},i=1,2,\ldots ,N\right\}$; mapping it to a high-dimensional feature space through the kernel function ψ, it has better aggregation and it can solve an optimal hyperplane in the feature space, so as to achieve the maximum separation between the target data, and the origin of the coordinates. The decision function is fsign(z)=sign(w×ψ(z)−ρ); it attempts to separate the training set from the origin, and maximize the distance between the hyperplane and the origin.

The weight w of the support vector and the threshold ρ can be described by solving the following quadratic programming problem:(4)$$\left\{ \begin{array}{l} min\frac{1}{2}w^{2}+\frac{1}{vN}\overset{N}{\underset{i=1}{\sum }}\xi_{i}-\rho \\ s.t.\;\left(w\times \psi \left(z\right)\right)\gg \rho -\xi_{i}\;\xi_{i}\geq 0 \end{array}\right.$$
where vϵ(0,1) is used to control the proportion of support vectors in the training samples. After introducing the kernel function, the above problem can be transformed into a dual problem:(5)$$\left\{ \begin{array}{l} min\frac{1}{2}\overset{N}{\underset{i=1}{\sum }}\overset{N}{\underset{j=1}{\sum }}\alpha_{i}\alpha_{j}k\left(z_{i},z_{j}\right)\\ s.t.\;0\leq \alpha_{i}\leq \frac{1}{vN}\;\overset{N}{\underset{i=1}{\sum }}\alpha_{i}=1 \end{array}\right.$$

In OCSVM, $\rho =\overset{N}{\underset{i=1}{\sum }}\alpha_{i}k(z_{i},z_{j})$ is a determined threshold, determining the separation hyperplane with the weight vector w, through the decision equation, OCSVM can determine whether the sample z is a fault sample [17].

### 2.4. Construction of the Hierarchical Classifier

Because the types of fault samples are not comprehensive, there are some unknown types of faults occurring in practical work. When an unknown type of fault occurs, by using multi-class classifier to identify HVCBs mechanical faults, the unknown faults will be identified as a known fault or normal. Although OCSVM can accurately monitor the state of mechanical failure, it cannot identify the type of fault as known or unknown. Therefore, a hybrid classifier is constructed with OCSVM and RF. Using OCSVM to avoid mistaken identifications of fault status, we further identified the unknown fault types accurately without training samples, through RF and OCSVM.

Figure 1 is a flowchart of fault diagnosis based on a hybrid classifier. Firstly, the OCSVM_0_ classifier is applied to identify the normal and fault states of HVCB. If the HVCB is in a fault condition, RF is used to identify the specific fault types. Then, aiming at the fault condition identified by RF, OCSVM*_l_* (where *l* is a three-fault condition) is used to identify and correct the condition, based on the OCSVM*_l_* model trained by a specific known fault type.

## 3. Feature Extraction of Original Vibration Signals Based on Time Domain Segmentation

### 3.1. Signal Acquisition System

The experiment was carried out on LW9-72.5 series SF6 HVCBs, using a spring control mechanism. The vibration signal acquisition system was built using a piezoelectric accelerometer and a NI 9234 data acquisition card produced by National Instruments. The system is as shown in Figure 2. The accelerometer was installed on the mechanism box near the operating mechanism. The position was close to the vibration source of the HVCBs, and could record the vibration more clearly, without affecting the performance of the circuit breaker. The sampling frequency was 25.6 kS/s. The coordinate origin of the sampled signal was the time when the circuit breaker would act (The trigger sends acquisition instructions to the data acquisition card). The recording starting point and the time of the acquisition signal is the same in the four conditions.

### 3.2. Feature Extraction Based on Time Domain Segmentation of the Original Signal

To extract the features of HVCBs vibration signals in a specific time period for the original signal, a uniform time scale was used to segment the original signal in the time domain, and the time domain features of each segment were extracted after segmentation. The time domain feature was extracted directly from the segmented signals, and a loss of high-frequency information could be avoided in time–frequency processing. The integrity of the feature information was guaranteed, and it saved time in signal processing.

In order to analyze the influences of features on the accuracy of fault diagnosis under different segmentation scales, from the original signal, it can be seen that the actions of the iron core stagnation fault were delayed, compared with the normal signal. The amplitude of the base screw loosening fault signal was smaller and the attenuation process was slow. The amplitude of the poor lubrication fault was relatively small. The time of the operating mechanism received trigger instructions to the vibration signal amplitude changes (Ts), and the amplitude reached a peak (Tp) to be the unit, respectively. The signal is divided into 29 segments and nine segments by Ts and Tp. Different fault types are segmented at the same scale. There are obvious differences between two different scales and different signals, so we used two scales for time domain segmentation.

Figure 3 is a map of the measured vibration signal and the time domain segmentation unit. It can be seen from the original signal that with the iron core stagnation action delay, compared with the normal condition, the amplitude of screw loosening condition was smaller, the attenuation process was slower, and the amplitude of the poor lubrication condition was relatively small. Therefore, in the time domain segmentation, it can be seen that there were differences between different types of fault signals in the same period. The features were extracted from each segmented signal, and used to identify the mechanical conditions of the HVCBs.

The new method extracts 17 times domain features that can reflect the amplitude changes and the attenuation degrees in different periods, and construct feature vectors [14,15,16]. Table 1 is the feature calculation formula, where $p_{n}$ is the probability density, n=1,2,⋯,N, *N* is the number of sample points per segment after the time domain segmentation. Table 2 is the distribution of the features with different time domain segmentation scales.

## 4. Comparison of Classification Effects of Different Feature Extraction Methods

In order to compare the effects of feature extraction in a new method, three signal processing methods of EMD, EEMD, and LMD were used to extract features, to compare them with the new method. Figure 4 is the result of signal decomposition and time domain segmentation by using EMD, EEMD, and LMD. The time domain segmentation scale was the same as the new method. EMD and EEMD decompose the signal into some intrinsic mode functions (IMFs). LMD decomposes the signal into multiple product functions (PFs) with instantaneous frequency.

When using different feature extraction methods, Table 3 shows the recognition accuracy without unknown types when using multiple classifiers and the original feature set dimension. *Di* is the features dimension, and *Ac* is the accuracy of the condition recognition.

As shown in Table 3, when the new method divided the signal into 29 segments and nine segments, the new method could identify three states effectively, while the traditional signal processing method had mistaken identifications. In this paper, the feature dimensions extracted by the new method were 153 and 493. Compared with the traditional signal method, the feature dimension was lower. Therefore, the new method feature extraction not only improved the accuracy of the state recognition, but also effectively reduced the complexity of the original feature set.

Figure 5 shows the time statistics for extracting features by direct time-domain segmentation and traditional signal processing methods. According to Figure 5, no matter the time domain segmentation method that is used, the new method without signal processing removed the signal processing, and extracted the time-domain features only for a single time series of the original signal.

The computation time of the new method was lower than that of the traditional methods, which were needed to extract features from multiple IMF or PF time series. The efficiency of the feature extraction was higher than EMD, LMD, and EEMD. When the sampling rate of the vibration signal to be analyzed is higher, the advantages of the new method will be more obvious.

## 5. Feature Selection

In the existing feature selection, when the wrapper method was used; combined with the particle swarm algorithm, the feature subset satisfying the classification accuracy rate is found according to the classifier effect, but the efficiency of the optimization is low. In actual work, the filter method receives more applications. Experiments are carried out according to the statistical results of the features [23,24,25].

RF is an ensemble learning method. Its Gini importance index takes the comprehensive contribution of features in different feature combinations into account, and analysis becomes more comprehensive. The features were in descending order according to the Gini importance, and SFS was carried out to obtain a better candidate feature set. After that, the classifier was constructed with different feature subsets, and the evaluation index of RFC was calculated. Finally, the best feature subset was determined, which was used to train the optimal classifier.

### 5.1. The Gini Importance of Time Domain Features

In order to reduce the complexity of the classifier, three classes of signals were used as the classification targets. The complete original feature set was used as the input feature vector to train the RF classifier, and all of the Gini importance were obtained. The importance of the original feature set constructed by two times domain segmentation methods is shown in Figure 6. As is evident in Figure 6, the Gini importance of the different features were quite different. Therefore, feature ordering could be carried out according to the Gini importance.

In order to prove the effectiveness of feature classification ability, based on Gini importance, Under two kinds of time domain segmentation scales, two groups were selected from the highest and lowest Gini importance of the features, respectively. A box plot was drawn to analyze the distribution of the features. According to the distribution of features, the classification ability of the features with different Gini importance was intuitively compared. According to the Gini importance, the best features of 29 segments are F450 and F62, and the worst features are F311 and F313; the best features of the nine segments are F31 and F34, and the worst features are F16 and F37.

As shown in Figure 7, the box diagram distribution was determined by 10 groups of typical fault samples. The distribution of the features with high Gini importance was centralized with no crossing. The degree of distinction between the different classes was high. The distribution of the features with low Gini importance was wide and overlapped. The degree of distinction between the different classes was low. The validity of the feature classification ability was verified, based on Gini importance.

### 5.2. Sequential Forward Selection Based on RFC

In the process of sequential forward selection, the optimal feature subset was determined by the classification accuracy of the feature subset, and the *J_F_* index of RFC.

The separation of features could be determined by the Fisher criterion in pattern recognition [25]:(6)$$J=\frac{tr(\sigma_{b})}{tr(\sigma_{w})}$$

Fisher’s criterion *J* is a measurement of the separability among all classes. If the J value of the feature set is larger for the training set, the diversity of this feature is better. Where $\sigma_{b}$ and $\sigma_{w}$ are the between-class scatter matrix and the within-class scatter matrix, respectively. The calculation formula is as follows:(7)$$\left\{ \begin{array}{l} \sigma_{b}=\sum_{i}^{c}n_{i}(m_{i}-m){\left(m_{i}-m\right)}^{T}\\ \sigma_{w}=\sum_{i}^{c}\frac{1}{n_{i}}\sum_{x\in w_{i}}\left(x-m_{i}\right){\left(x-m_{i}\right)}^{T} \end{array}\right.$$

The linear transformation matrix *W* transforms the Fisher criterion to [26]:(8)$$J_{F}(W)=tr\left(\frac{W^{T}\sigma_{b}W}{W^{T}\sigma_{w}W}\right)$$

When $\sigma_{W}$ is singular or ill-conditioned, a diagonal matrix λI with λ>0 is added to $\sigma_{w}$. Since $\sigma_{w}$ is symmetric positive semi-definite, $\sigma_{w}+\lambda I$ is non-singular with any λ>0.

To overcome this shortcoming, the RFC was adopted in the new method. The classification effect of different feature sets was analyzed. By replacing the regularized matrix $\sigma_{w}$ in (8), the RFC becomes [26]:(9)$$J_{F}(W)=tr(\frac{W^{T}\sigma_{b}W}{W^{T}(\sigma_{w}+\lambda I)W})$$

Therefore, the problem of singularity is solved, and it can be applied in our feature selection algorithm to measure the classification ability of different features [26].

In the condition of time domain segmentation with 29 and nine segments, respectively, the 50 most important dimensional features were arranged in descending order, and the SFS method was used to construct different feature subsets. Figure 8 is the RF classification accuracy, and *J_F_* is the feature selection process.

As it can be seen from Figure 8, the accuracy of the feature subset dimension of the two segmentation scales was 100% when the feature subset dimensions were nine- and 10-dimensional, which cannot measure the effect from a single accuracy. With the increase in features, the evaluation index *J_F_* of the feature subset first increased and then decreased. Finally, the maximum *J_F_* value was used to determine the final feature subset. The best feature subset dimension were 12 and 33, when time domain segmentation scales were 29 and nine segments, respectively. At this time, when the time domain was divided into 29 segments and nine segments in the time domain, *J_F_* reached the maximum value. When classifying with this feature subset, the class separability was the highest. Therefore, *J_F_* and the feature dimension were considered comprehensively, and a classifier model was constructed, based on the best feature subset, with 12 dimensions of 29 segments. The best subsets of the features are shown in Table 4.

## 6. Analysis of the Recognition Effect of the Hybrid Classifier with Unknown Faults

After determining the optimal subset of features, this paper designed an identification experiment with an unknown fault type, and verified the advantages of the hybrid classifier adopted by the new method. The screw loosening condition in the experiment was regarded as an unknown fault of the untrained sample, and it only participates in the final test without participating in the training process of the classifier. In the experiment, the error limit v and RBF kernel width parameter were 0.82 and 17.68, respectively [27].

In order to compare the classification effects of multiple classifiers, RF and SVM were used to analyze normal and three fault types (the iron core stagnation, the screw loosening condition, and the poor lubrication condition) with untrained samples. The classifier to build the best feature subset was determined by the new method. SVM parameter reference [9] setting. Among them, the screw looseness was regarded as an unknown fault state without training samples, and it participated in two classifiers but it did not participate in training. Twenty groups of normal samples of two known fault states (without an unknown type of fault) were used to train the classifier. Ten groups of normal samples of three fault states (with an unknown type of fault) were used to test the classifier. The classification results are shown in Table 5.

From the results of Table 5, RF and SVM accurately identified the fault types of the HVCBs. Neither RF nor SVM could identify an unknown fault type accurately (C3), in the state recognition without training samples. RF identified one group of samples as normal, four groups of samples were identified as iron core stagnation, and the five groups are identified as having poor lubrication condition, while SVM identified 10 groups of samples as being normal. The reliability of SVM was low. Therefore, the RF classifier has advantages, and its classification is more reliable.

In order to prove that the new method of the hierarchical hybrid classifier is used to identify the unknown type without training samples, a comparative test was carried out between the new method and the OCSVM-RF method. Table 6 is the result of two hybrid classifiers for OCSVM-RF (O-R) and the new method OCSVM-RF-OCSVM (O-R-O). In contrast, O-R could make up for the shortcoming that RF mistakenly identified an unknown condition as a normal condition, but the 10 untrained samples were wrongly identified as known fault types.

## 7. Conclusions

This paper proposes a novel mechanical fault feature selection and diagnosis approach for HVCBs, using features extracted without signal processing. The signal is not processed by digital signal processing methods, and it extracts features directly in the time domain. Feature selection is used to gain the best feature subsets for achieving high efficiencies and accuracies of fault recognition. The main contribution of the new approach is as follows:(1)The features is extracted directly after the time domain segmentation of the original signal, with less time complexity.(2)Feature selection is adopted to reduce the optimal feature subset dimension, the time consuming nature of feature extraction, and the complexity of the classifier model.(3)The hierarchical hybrid classifier avoids the limitation of identifying the fault samples as the normal condition, and identifies the unknown fault types effectively. Compared with the traditional multiple classifiers, the condition recognition effect is improved.

HVCB has many kinds of faults, degrees of faults. It is difficult to obtain all the data that is needed for relevant experiments. More samples of fault types and degree data for further experimental study will be accumulated in future work.

## Figures and Tables

**Figure 1 sensors-19-00288-f001:**
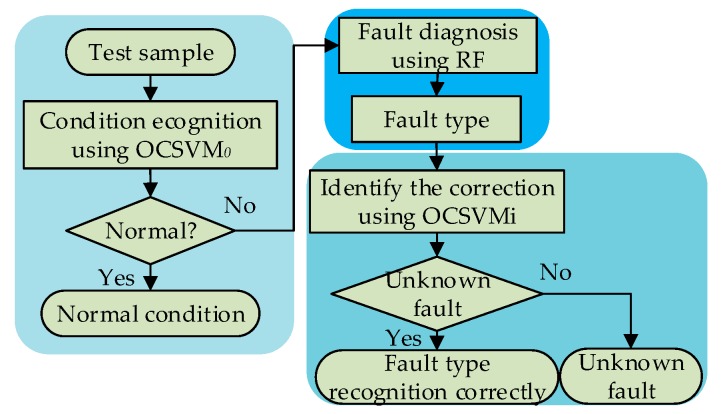
Diagnosis process of the hybrid classifier.

**Figure 2 sensors-19-00288-f002:**
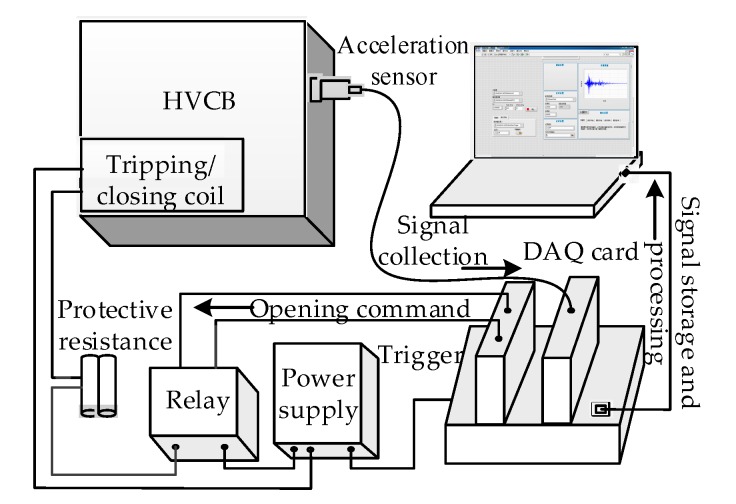
Vibration signal acquisition system.

**Figure 3 sensors-19-00288-f003:**
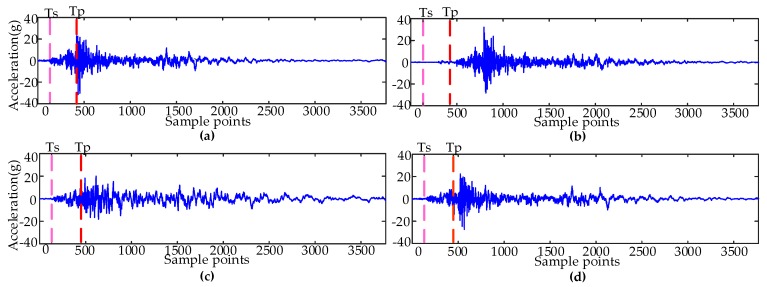
Measured vibration signals and the time domain segmentation unit: (**a**) The normal condition; (**b**) The iron core stagnation; (**c**) The screw loosening condition; (**d**) The poor lubrication condition.

**Figure 4 sensors-19-00288-f004:**
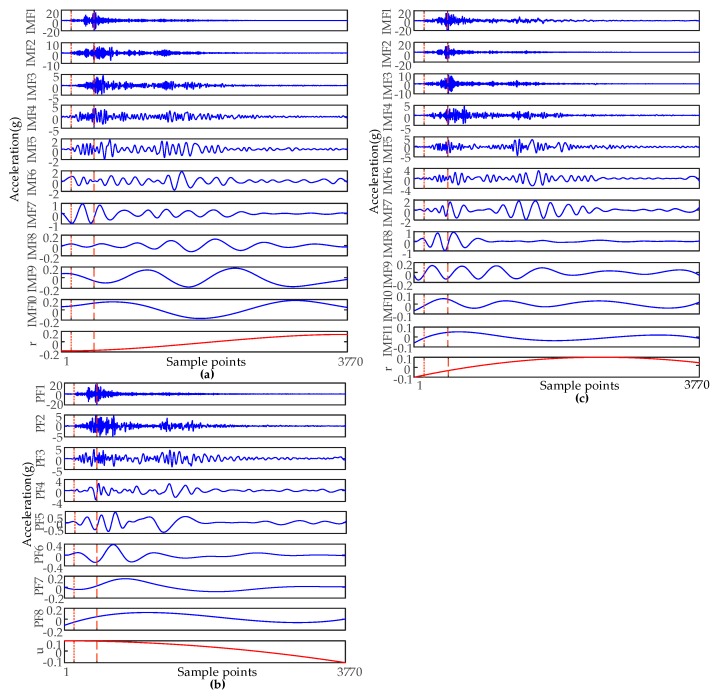
The time domain segmentation of EMD, LMD, and EEMD: (**a**) IMFs decomposed by EMD method; (**b**) PFs decomposed by the LMD method; (**c**) IMFs decomposed by the EEMD method.

**Figure 5 sensors-19-00288-f005:**
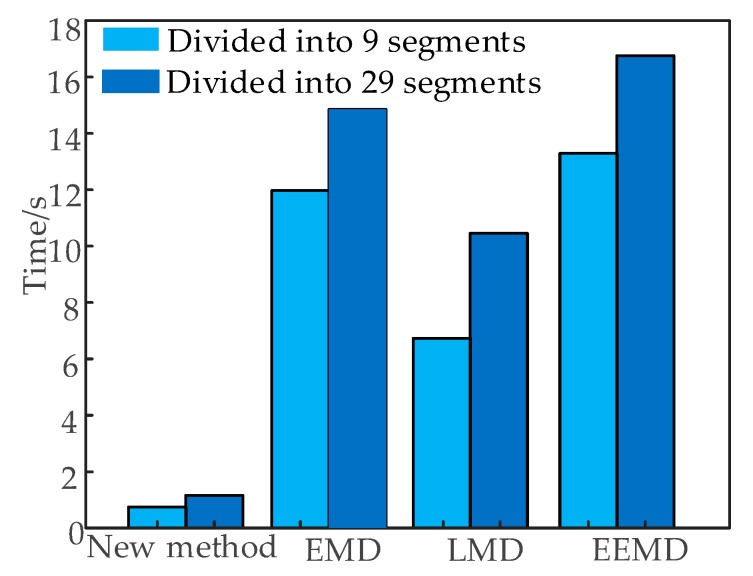
Time statistics of the feature extraction.

**Figure 6 sensors-19-00288-f006:**
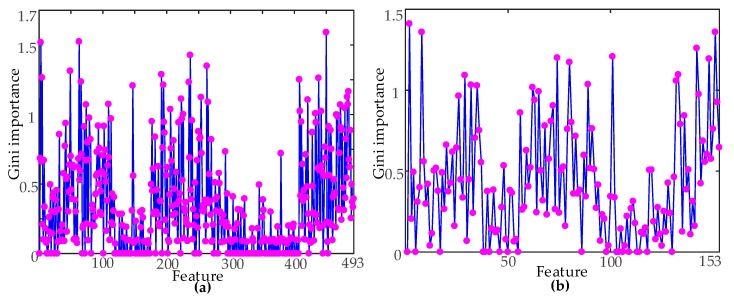
Gini importance: (**a**) Gini importance of 29 segments in time domain segmentation; (**b**) Gini importance of nine segments in time domain segmentation.

**Figure 7 sensors-19-00288-f007:**
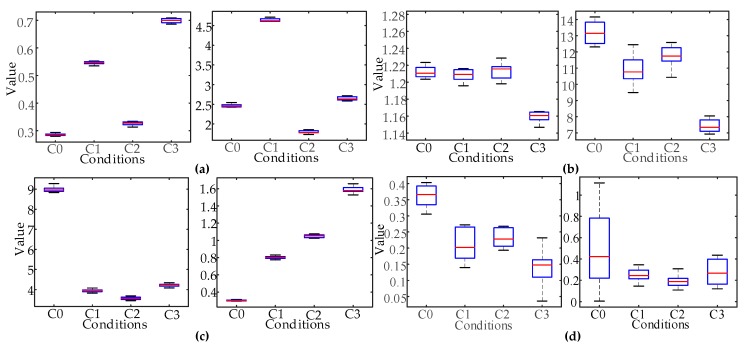
Feature distribution between high and low Gini importance: (**a**) 29 segments with the highest Gini importance; (**b**) 29 segments with the lowest Gini importance; (**c**) nine segments with the highest Gini importance; (**d**) nine segments with the lowest Gini importance.

**Figure 8 sensors-19-00288-f008:**
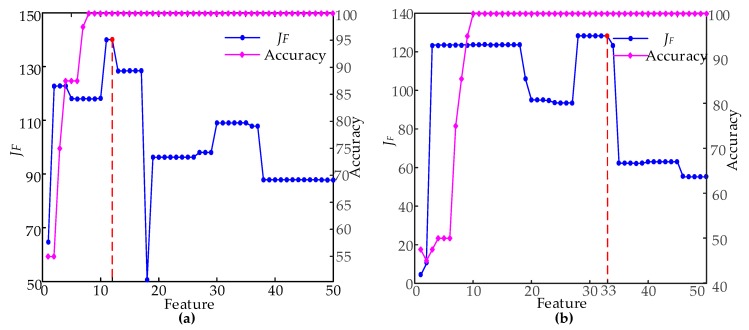
*J_F_* and classification accuracy of different feature sets: (**a**) 29 segments of time domain segmentation; (**b**) nine segments of time domain segmentation.

**Table 1 sensors-19-00288-t001:** Formula of features.

Feature	Formula	Feature	Formula
Mean value	$F_{mv}=\frac{1}{N}\sum_{n=1}^{N}x(n)$	Standard deviation	$F_{std}=\sqrt{\frac{1}{N}\sum_{n=1}^{N}{\left(x(n)-F_{mv}\right)}^{2}}$
Variance	$F_{tv}=\frac{1}{N}\sum_{n=1}^{N}{\left(x(n)-F_{mv}\right)}^{2}$	Skewness	$F_{sv}=\frac{1}{N}\sum_{n=1}^{N}{\left(\frac{x(n)-F_{mv}}{F_{std}}\right)}^{3}$
Kurtosis	$F_{kv}=\frac{1}{N}{\sum_{n=1}^{N}\left(\frac{x(n)-F_{mv}}{F_{std}}\right)}^{4}$	Peak-to-peak value	$F_{ppv}=\max(x(n))-\min(x(n))$
Square root of amplitude	$F_{sta}={\left(\frac{1}{N}\sum_{n=1}^{N}\sqrt{\left|x(n)\right|}\right)}^{2}$	Mean amplitude	$F_{av}=\frac{1}{N}\sum_{n=1}^{N}\left|x(n)\right|$
Peak value	$F_{pv}=\max\left(\left|x(n)\right|\right)$	Shape factor	$F_{sf}=\frac{F_{rms}}{F_{av}}$
Crest factor	$F_{cf}=\frac{F_{pv}}{F_{rms}}$	Impulse factor	$F_{if}=\frac{F_{pv}}{F_{av}}$
Margin factor	$F_{mf}=\frac{F_{pv}}{F_{sra}}$	Shannon entropy	$F_{se}=-K\overset{N}{\underset{n=1}{\sum }}p_{n}\log p_{n}$
Renyi entropy	$F_{re}=\frac{1}{1-\alpha }\log\overset{N}{\underset{n=1}{\sum }}p_{n}{}^{\alpha }$	Tsallis entropy	$F_{te}=-\frac{1}{\alpha -1}\log(1-\overset{N}{\underset{n=1}{\sum }}p_{n}{}^{\alpha })$
Root-mean-square value	$F_{rms}={\left(\frac{1}{N}\sum_{n=1}^{N}x{\left(n\right)}^{2}\right)}^{1/2}$		

**Table 2 sensors-19-00288-t002:** Distribution of features.

Feature	Feature Number (Ts)	Feature Number (Tp)	Feature	Feature Number (Ts)	Feature Number (Tp)
Mean value	F1–F29	F1–F9	Standard deviation	F30–F58	F10–F18
Variance	F59–F87	F19–F27	Skewness	F88–F116	F28–F36
Kurtosis	F117–F145	F37–F45	Peak-to-peak value	F146–F174	F46–F54
Square root of amplitude	F175–F203	F55–F63	Mean amplitude	F204–F232	F64–F72
Peak value	F233–F261	F73–F81	Shape factor	F262–F290	F82–F90
Crest factor	F291–F319	F91–F99	Impulse facto	F320–F348	F100–F108
Margin factor	F349–F377	F109–F117	Shannon entropy	F378–F406	F118–F126
Renyi entropy	F407–F435	F127–F135	Tsallis entropy	F436–F464	F136–F144
Root-mean-square value	F465–F493	F145–F153			

**Table 3 sensors-19-00288-t003:** State recognition results.

Test	9 Segments					29 Segments				
	C1	C2	C3	Di	Ac (%)	C1	C2	C3	Di	Ac (%)
C1	10	0	0	153	100	10	0	0	493	100
C2	0	10	0	153	100	0	10	0	439	100
C3	0	0	10	153	100	0	0	10	439	100
EMD-C1	10	0	0	1530	100	10	0	0	4930	100
EMD-C2	1	9	0	1530	90	0	10	0	4390	100
EMD-C3	0	0	10	1530	100	1	0	9	4390	90
LMD-C1	10	0	0	1071	100	10	0	0	3451	100
LMD-C2	0	10	0	1071	100	0	9	1	3451	90
LMD-C3	1	0	9	11	90	0	0	10	3451	100
EEMD-C1	10	0	0	1377	100	10	0	0	4437	100
EEMD-C2	0	10	0	1377	100	0	10	0	4437	100
EEMD-C3	0	2	8	1377	80	0	1	9	4437	90

**Table 4 sensors-19-00288-t004:** The best subset of features.

Feature Number	Feature Description
F450	Mean amplitude of 27 segments
F62	Peak value of four segments
F2	Standard deviation of one segment
F236	Renyi entropy of 14 segments
F262	Square root of the amplitude of 16 segments
F48	Shannon entropy of three segments
F191	Skewness of 12 segments
F4	Skewness of one segment
F438	Margin factor of 26 segments
F408	Root-mean-square value of 24 segments
F65	Shannon entropy of four segments
F234	Margin factor of 14 segments

**Table 5 sensors-19-00288-t005:** Diagnosis result of the patterns not contained in the training samples, using random forest (RF) and support vector machine (SVM).

Test	RF					SVM				
	C0	C1	C2	C3	Ac	C0	C1	C2	C3	Ac
C0	10	0	0	0	100	10	0	0	0	100
C1	0	10	0	0	100	0	10	0	0	100
C2	0	0	10	0	100	0	0	10	0	100
C3	1	4	5	0	0	10	0	0	0	0

**Table 6 sensors-19-00288-t006:** Diagnosis result of the patterns not contained in the training samples using the O-R and O-R-O.

Test	O-R					O-R-O				
	C0	C1	C2	C3	Ac	C0	C1	C2	C3	Ac
C0	10	0	0	0	100	10	0	0	0	100
C1	0	10	0	0	100	0	10	0	0	100
C2	0	0	10	0	100	0	0	10	0	100
C3	0	4	6	0	0	0	0	0	10	100

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
