# Peer review of "A Novel Mechanical Fault Feature Selection and Diagnosis Approach for High-Voltage Circuit Breakers Using Features Extracted without Signal Processing"

_sensors, 2019, doi:10.3390/s19020288_

Round 1

Reviewer 1 Report

I would like to thank the effort done by the authors. However, in my opinion, I didn't find a new contribution as compared to previous works related to diagnosis for high voltage circuit breaker. The proposed method is compared against other methods that are already know that have several issues such as EMD and EEMD.

This reviewer suggests to use the proposed method in the identification of multiple single faults at the same time that could be present in some electric devices. Because the identification of single faults in HVCB is already done.

The literature review in Introduction needs completely rewritten. The material is organized very messy, it does not reflect the current status in this field, especially in the feature extraction based fault diagnosis area.

Author Response

Dear Reviewer,

Thank you very much for your insightful comments concerning our previous manuscript (ID: sensors-406048). These comments are all valuable and very helpful for improving the quality of our work. We have done some modifications based on our previous manuscript according to your comments, and resubmitted this manuscript. The detail responses are in the following file. 

Reviewer 2 Report

Comments on “A novel mechanical fault feature selection and diagnosis approach for high voltage circuit breaker using features extracted without signal processing”

General overview

·         I believe this article is ready for publication, but there are some minor issues or improvements that can be done.

·         The title could change to “…for a high voltage circuit breaker …” to be expressed as a particular case, or otherwise it could be generalized by “…for high voltage circuit breakers …”.

·         There are some lines that are confusing or lack the right article.

·         The abstract has some room for improvement.

Comments on the Abstract

Line 17: The term HVCB is used, but is not declared in the abstract, this must be declared as High voltage circuit breaker, in both the abstract and the rest of the article.

Line 18: this line from  “… by time scale…” onwards is not clear, maybe it should be re-writen.

Line 20: “…for constructing a vector…”, also could it be “Adopting the ensemble learning method”

Line 23: “…(RFC) is used…”

Other minor corrections

Line 33: could it be “control and primary protection…”.

Line 35: “…of the power system.”.

Line 71: “….all the fault types…”.

Line 79: “and use it, to construct the feature vector.”.

Line 83: “by the measured signal”

Line 86: Perhaps way could be replaced by method, this is more formal in my opinion.

Line 122: well could be replaced by properly, because well leads to the question, well respect what?.

Line 141: There is an “’” in the middle of OCSVM.

Line 144: The expression to occur in practical work, is confusing, the point is not clear,. I recommend to re-write that sentence.

Line 183: “…of the measured vibration signal and the time domain...”

Line249: “…the tree classes/types of signal are used as…”

Line 253: “…importance of the different …”

In general, the paper seems very interesting to me.

Author Response

(The authors gave the same response as above.)

Reviewer 3 Report

This paper presents an algorithm that can determine the fault feature of HVCB mainly using hybrid classifier of SVM-RF. It is worth to be considered for the publication because of the logical composition as a whole.

1. Abstract should be balanced including the problems of the existing method, the verification of the results of the research, the problems, and future development direction rather than the detailed development process.

2. In line 54-58, there is more information and features in the frequency domain than the time domain. Please explain and add this issue.

3. please add proper references to section 2.1 Gini importance

4. In line 124, 2.2 OCSVM -> 2.2 one-class support vector machine (OCSVM)

5. Why did you set the sampling rate at 25.6 kS/s? What happens when the samping rate is increased or decreased?

6. Using proper analog or digital filters is expected to produce better results. Need explanation?

7. Please add more explanations to the measurement position of the accelerometer?

8. In Fig. 3, add more explanations of Ts and Tp. Ts seems to be a starting point for the signal, but Tp could not be understood.

9. How can you determine if the four types of signals in Fig. 3 are representative? time averaging of signals, the enough number of signals?

10. How can you determine 17 features in Table 1 for the analysis?

11. In Table 3, there is a lack of explanation for the reasons for division into 9 and 29 segments.

12. What is v and σ in line 316?

13. The conclusion should describe the advantages and disadvantages of this paper, the verification part, the limitations, and the future research.

Author Response

(The authors gave the same response as above.)

Round 2

Reviewer 1 Report

After the revision of this new manuscript and the author’s responses; this reviewer can conclude the new submission has been significantly improved from the first version, and most of my concerns were clarified adequately.

Therefore, the new submitted article has been done well. Authors have addressed all issues.

Thus, this reviewer recommends this manuscript for publication in Sensors.

Author Response

Dear Reviewer,

Thank you very much for your insightful comments concerning our previous manuscript . These comments are all valuable and very helpful for improving the quality of our work. 

We have added the relevant literatures as follows.

[2] Wan, ST.; Chen, L.; Dou, LJ.; Zhou, JP. Mechanical Fault Diagnosis of HVCBs Based                on Multi-Feature Entropy Fusion and Hybrid Classifier, Entropy 2018, 20, 847.

[9] Bustos, A.; Rubio, H.; Castejon, C.; Garcia-Prada, JC. EMD-Based Methodology for the Identification of a High-Speed Train Running in a Gear Operating State, Sensors 2018, 18, 793.

[21] Ma, S.; Chen, M.; Wu, J.; Wang, Y.; Jia, B.; Yuan, J. Intelligent Fault Diagnosis of HVCB with FeatureSpace Optimization-Based Random Forest, Sensors 2018, 18, 1221.[24] Dang, Z.; Lv, Y.; Li, YR.; Wei, GQ. Improved Dynamic Mode Decomposition and Its Application to Fault Diagnosis of Rolling Bearing, Sensors 2018, 18, 1972.